# Universal Tick Vaccines: Candidates and Remaining Challenges

**DOI:** 10.3390/ani13122031

**Published:** 2023-06-19

**Authors:** Luís Fernando Parizi, Naftaly Wang’ombe Githaka, Carlos Logullo, Jinlin Zhou, Misao Onuma, Carlos Termignoni, Itabajara da Silva Vaz

**Affiliations:** 1Centro de Biotecnologia, Universidade Federal do Rio Grande do Sul, Porto Alegre 91501-970, Brazil; luisfparizi@gmail.com (L.F.P.); carlos.termignoni@ufrgs.br (C.T.); 2International Livestock Research Institute (ILRI), P.O. Box 30709, Nairobi 00100, Kenya; n.githaka@cgiar.org; 3Instituto de Bioquímica Médica Leopoldo de Meis, Universidade Federal do Rio de Janeiro, Rio de Janeiro 21941-853, Brazil; carlos.logullo@bioqmed.ufrj.br; 4Instituto Nacional de Ciência e Tecnologia em Entomologia Molecular, Universidade Federal do Rio de Janeiro, Rio de Janeiro 21941-853, Brazil; 5Key Laboratory of Animal Parasitology of Ministry of Agriculture, Shanghai Veterinary Research Institute, Chinese Academy of Agricultural Sciences, Shanghai 200241, China; jinlinzhou@shvri.ac.cn; 6Department of Infectious Diseases, Graduate School of Veterinary Medicine, Hokkaido University, Sapporo 060-0818, Japan; 7Departamento de Bioquímica, Universidade Federal do Rio Grande do Sul, Porto Alegre 90040-060, Brazil; 8Faculdade de Veterinária, Universidade Federal do Rio Grande do Sul, Porto Alegre 91540-000, Brazil

**Keywords:** cross-protection, parasite, tick, tick-control, vaccine

## Abstract

**Simple Summary:**

Various new technologies have been used to learn more about tick physiology and biology in hopes of developing vaccines to protect people and animals from tick infestations. However, creating these vaccines has been harder than expected. Even though early tests showed promise, the development of new commercial anti-tick vaccines has not been successful. Since various tick molecules have redundant or complementary functions, it is necessary to select more than one molecule to include in the vaccine. Additionally, ticks are spreading to new areas and affecting new animal and human populations, impelling urgency to find new control methods. This review focuses on the challenges and progress made in recent years in developing vaccines against different tick species.

**Abstract:**

Recent advancements in molecular biology, particularly regarding massively parallel sequencing technologies, have enabled scientists to gain more insight into the physiology of ticks. While there has been progress in identifying tick proteins and the pathways they are involved in, the specificities of tick-host interaction at the molecular level are not yet fully understood. Indeed, the development of effective commercial tick vaccines has been slower than expected. While omics studies have pointed to some potential vaccine immunogens, selecting suitable antigens for a multi-antigenic vaccine is very complex due to the participation of redundant molecules in biological pathways. The expansion of ticks and their pathogens into new territories and exposure to new hosts makes it necessary to evaluate vaccine efficacy in unusual and non-domestic host species. This situation makes ticks and tick-borne diseases an increasing threat to animal and human health globally, demanding an urgent availability of vaccines against multiple tick species and their pathogens. This review discusses the challenges and advancements in the search for universal tick vaccines, including promising new antigen candidates, and indicates future directions in this crucial research field.

## 1. Introduction

Veterinary ectoparasites, such as ticks, cause enormous economic losses, both directly through bloodsucking and irritation that affect animal welfare, and indirectly through the transmission of often debilitating diseases such as babesiosis, theileriosis, and anaplasmosis [1]. Besides the high cost of currently available control methods, the increasing resistance to the drugs used to control vector infestation and treatment for tick-borne diseases is a major concern worldwide [2]. To enhance the introduction of new animal health products into specific markets, such as sub-Saharan Africa or South America, it is crucial to implement innovative strategies in product development that target the diseases endemic in these regions [3].

Vaccination is usually proposed as a safe and sustainable strategy to overcome problems related to tick infestation and disease transmission. However, progress in anti-vector vaccine development has been slow and patchy, and currently, only a handful of vaccines targeting ectoparasites have been developed and tested successfully [4,5]. Moreover, the commercialization of promising vaccine candidates for livestock disease is still a complicated process despite lower regulatory thresholds compared to human vaccines [6,7]. Regarding vector-borne diseases endemic in low-income countries, it is difficult to recoup investments in vaccine development in the absence of government support [8].

The evolutionary complexity of parasites and their intimate relationships with their vertebrate hosts pose a daunting challenge to rationally design vaccines against these organisms [9]. All of them can trigger host innate and adaptive immunity that is often stage-specific [10]. However, parasites (including ticks) possess potent defense mechanisms to overcome or circumvent host immune responses, thus re-establishing the balance in favor of the parasite [11,12,13,14,15,16,17]. Tick-host interactions over the past 100 million years resulted in the coevolution of an enormous range of proteins and non-protein factors secreted in tick saliva. These factors are used initially to locate their hosts and feeding sites and subsequently to inhibit or modulate the host defense [18], including the hemostatic and immune system [19,20,21].

On the other hand, some host species promptly develop protective immune responses that reject ticks. Therefore, understanding the mechanisms that trigger this protection is crucial in order to gain a better understanding of tick-host interaction and to find molecular targets for a vaccine [22]. In some cases, hosts are able to develop a boosting protection after a second tick infestation [23]. However, due to economic and health reasons, it is not feasible to rely on the development of natural immunity in order to achieve a reasonable level of protection for humans and animals against ticks. Moreover, since ticks are vectors of many important pathogens [24], it is critical to develop a preventive protection method before the onset of parasitism [25].

Universal vaccines against ticks, i.e., vaccines able to protect against different tick species, could reduce the costs of control programs. This is because hosts infested by two or more tick species in one area could be protected using a single vaccine [26]. This advantage encouraged many research groups worldwide to analyze tick proteins that were originally used as antigens for homologous vaccination [27] in trials against several tick species. Previously, several tick proteins were suggested as candidates for a universal vaccine [28,29]. This review provides updated information on antigen candidates on cross-vaccination trials that showed protection against tick infestation performed since then, shedding light on methodological advances in the search for protective antigens. Moreover, we have addressed remaining challenges in developing universal anti-tick vaccines, rational strategies for improving tick vaccines, and the impact of tick expansion into new territories on vaccine development.

## 2. Previously Described Universal Vaccine Antigen Candidates

### 2.1. Bm86

Bm86 protein is a *Rhipicephalus (Boophilus) microplus* (Canestrini, 1888) structural gut glycoprotein that was identified after a series of sequential fractionations of tick proteins, which were used to protect bovines against tick infestations [30,31]. This research was seminal because it introduced the concept of concealed antigen, defined as an antigen that is not encountered by the host immune system under natural conditions but can be reached by host antibodies in the case the host was previously immunized with this molecule. Recombinant proteins based on *R. (B.) microplus* Bm86 are the only antigens used in commercially developed anti-tick vaccines ever developed, named TickGard [32] and Gavac [33]. Nowadays, however, only Gavac is commercially available in some regions [34]. It has a 55–100% efficacy against *R. (B.) microplus* populations in Latin America [35]. Since Bm86 discovery in the 1980s, huge efforts were undertaken to find protective antigens to be included in universal vaccines [29]. Preliminary results showed that Bm86 and other tick proteins induce partial cross-protection levels, leading many research groups to perform vaccination trials testing putative universal antigens. Table 1 summarizes such trials in recent years.

Previously, an overall 50% reduction in the engorgement weight of *H. a. anatolicum* nymphs fed on Bm86-vaccinated calves was obtained [52]. Furthermore, Bm86 and its orthologue Haa86 from *H. a. anatolicum* were used in a vaccination trial against *R. (B.) microplus* and *H. a. anatolicum* infestations [53]. Although both Bm86 and Haa86 presented cross-protection, these antigens showed a higher species-specific efficacy, similar to other *H. a. anatolicum* subolesin, calreticulin, and cathepsin L-like cysteine proteinase antigens [38]. Additionally, Bm86 was effective against *H. dromedarii* infestations in cattle and camels. On the other hand, the same antigen failed to protect cattle against *Amblyomma sculptum* (Berlese, 1888) [26]. In another cross-vaccination trial against *Hyalomma* spp., Hd86 (*Hyalomma scupense* (Schulze, 1919) Bm86 orthologue), immunized calves presented a decrease in the number of *H. scupense* larva, an effect not observed in calves that were immunized with Bm86 [54]. Intriguingly, Hd86 and Bm86 immunization increased *H. scupense* and *Hyalomma excavatum* (Koch, 1844) adult female weight, respectively, showing that Hd86 and Bm86 efficacy varies according to the tick life stage. It was suggested that Hd86 and Bm86 protection rates variation could result from differences in amino acid sequences of these antigens among tick species [55] and tick populations [56], as well as different levels of protein expression in immature and adult life stages [57].

Distinct Bm86 vaccine efficacies against *Rhipicephalus (Boophilus) annulatus* (Say, 1821) and *R. (B.) microplus* could also be related to the activity of tick endogenous protein degradation machinery (proteases) [58]. Since Bm86 protein expression levels in both tick species are similar, distinct physiologic factors could have caused an increased *R. (B.) annulatus* susceptibility to Bm86 vaccine. Proteomic analysis showed *R. (B.) annulatus* has reduced protein degradation machinery compared to *R. (B.) microplus*, which leads to a higher susceptibility to host immune response factors such as antibodies and the complement system [58]. It was proposed that Bm86 could be useful in controlling *R. (B.) annulatus* and *R. (B.) microplus* in endemic regions, e.g., North America, using vaccination and standard eradication practices simultaneously [59,60].

### 2.2. Subolesin

Subolesin and their orthologs (akirins) are transcription factors that regulate the expression of signal transduction and innate immune response genes in ticks and other parasites [61,62,63,64]. Cross-protection experiments showed high vaccine efficacies using subolesin and akirins against different parasite species, making these antigens very promising for a broad universal vaccine [29]. These trials supported further investigations analyzing the efficacy of subolesin antigens against different parasite and host species using alternative vaccination protocols. Mono- and multi-antigenic formulations of *R. appendiculatus*, *R. (B.) decoloratus* and *A. variegatum* subolesins were used in *B. indicus* and *B. indicus* × *B. taurus* crossbred cattle, with an efficacy range of 47–94% [39]. However, the protection developed by bovines immunized with this multi-antigenic formulation was not improved when compared to mono-antigenic formulations. A different vaccine strategy, using an oral vaccine combining *R. (B.) decoloratus* subolesin and heat-inactivated *Mycobacterium bovis*, induced protection against an *R. (B.) decoloratus* and *R. appendiculatus* infestation in *B. indicus* × *B. taurus* crossbred cattle [41].

In light of evidence showing that subolesin and akirin induce partial protection against *Hyalomma* spp. and *Rhipicephalus* spp. infestations in red and white-tailed deer [65], field vaccination trials were performed using roe deer, another cervid that is highly important as a host for ticks and tick-borne pathogens in Europe [66]. For this trial, a chimera of *Ixodes scapularis* (Say, 1821) subolesin and *Aedes albopictus* (Skuse, 1894) mosquito akirin (Q38) was used, since its protective epitopes are conserved among distinct parasites [67], which is a necessary feature for a universal vaccine. Although the tick species collected in the field were not definitely identified, the roe deer showed a smaller number of ticks (likely *Hyalomma marginatum* (Koch, 1844), *Hyalomma lusitanicum* (Koch, 1844), *Rhipicephalus bursa* (Canestrini and Fanzago, 1878), and *Ixodes ricinus* (Linnaeus, 1758) compared with non-vaccinated animals. Noteworthy, Q38 was able to induce immune protection in rabbits against larvae of *I. ricinus* and *Dermacentor reticulatus* (Fabricius, 1794) [68], showing Q38’s potential to protect against different tick life stages.

Vaccinations trials using subolesin against the soft ticks *O. erraticus* and *O. moubata* showed a low degree of protection [69], probably due to the induction of non-protective antibodies for immunodominant subolesin epitopes in highly structured regions [37]. Subsequently, the prediction of subolesin linear B-cell epitopes, using peptide arrays, epitope mapping (pepscan), and modelling, was used to design synthetic peptides that increased the protection rates against these ticks [37]. Besides the use of the complete protein as an antigen, immunization with synthetic subolesin peptides stimulated the host immune system to develop antibodies against subolesin unstructured regions. Therefore, the prediction of B-cell and T-cell epitopes represents a promising tool to improve the immunogenicity of other tick antigen candidates that show low degree of protection [70], as well as to design chimeras containing subolesin or other protective antigenic epitopes [71]. Recently, an antigenic B-cell epitope peptide from subolesin was tested as a potential vaccine candidate using artificial capillary feeding experiments [72] and vaccination under stable [73] and field [74] conditions, showing partial protection against tick infestation.

Moreover, an alternative approach using a priming immunization with a DNA vaccine expressing *R. haemaphysaloides* subolesin, followed by boosters of *R. haemaphysaloides* recombinant subolesin or *R. haemaphysaloides* subolesin and *H. longicornis* P0 chimeric polypeptide provided cross-protection against *H. longicornis* of 79% and 86%, respectively [40]. The combination of a DNA vaccine with mono- or multi-protein immunizations is a promising strategy to increase the efficacy of anti-tick vaccines.

### 2.3. Glutathione S-Transferases

Glutathione S-transferases (GSTs) are enzymes involved in the detoxification of xenobiotics and endogenous compounds [75]. The potential of *H. longicornis* GST (GST-Hl) to protect bovines against *R. (B.) microplus* parasitism has been analyzed [76]. A vaccination trial showed that GST-Hl combined with *R. (B.) microplus* Vitellin-Degrading Cysteine Endopeptidase and *Boophilus* Yolk pro-Cathepsin decreased *R. (B.) microplus* infestation in cattle, conferring an overall protection ranging from 35% to 61% for 3 months [76]. Moreover, *R. appendiculatus* adult females fed on rabbits immunized with GST-Hl are negatively affected, leading to an overall vaccine efficacy of 67% [42]. However, the same antigen was ineffective in protecting rabbits against *R. sanguineus* s.l. [42]. Furthermore, immunogenicity and cross-recognition of GSTs from *R. appendiculatus*, *R. (B.) decoloratus*, *R. (B.) microplus*, *A. variegatum*, and *H. longicornis* were analyzed [43]. *Rhipicephalus (B.) decoloratus* and *A. variegatum* GST presented high cross-protection potential and were selected to compose an experimental cocktail vaccine, which was able to reduce 35% of *R. sanguineus* s.l. adult female numbers in immunized rabbit. All together, these data demonstrate that GSTs are potentially useful antigens to compose vaccines against *Rhipicephalus* spp.

## 3. New Cross-Protection Antigen Candidates

Protein P0, one of the eukaryotic 60S ribosomal proteins, has a high identity among ticks of different genera but a low sequence similarity with their hosts’ P0 [77], which is an important feature for candidate antigens. *Rhipicephalus sanguineus* s.l. and *R. (B.) microplus* P0 have identical sequences, from which a peptide composed of 20 amino acids within a highly immunogenic region was synthetized and used in vaccination trials [77]. The overall efficacy of this peptide as a protective antigen was 90% for *R. sanguineus* s.l. [77] and 54% for *A. mixtum* [45] in rabbits, and 96% against *R. (B.) microplus* in cattle [44]. Surprisingly, when the same peptide was tested as a conjugate with Bm86 to immunize dogs and cattle, the efficacies were 95% (peptide alone) and 85% (conjugated) against *R. sanguineus* s.l., and 89% (peptide alone) and 84% (conjugated) against *R. (B.) microplus* [78], indicating that the effect is not additive. However, *O. erraticus* P0 showed no cross-protection against *O. moubata* infestation [50]. Thus, the antigen P0 may be useful to compose a broad-spectrum vaccine against hard ticks, as well as against pathogens transmitted during tick feeding, as demonstrated for *Babesia* spp. [79].

Using a vaccinomic approach, new tick proteins with high efficacy in controlling *I. ricinus* and *D. reticulatus* in rabbits and dogs [80] were successfully identified. These trials showed efficacies up to 80% in animals individually inoculated with the following proteins: (1) heme lipoprotein from *I. ricinus*, (2) glypican-like protein, (3) secreted protein involved in homophilic cell adhesion, (4) sulfate/anion exchanger, or (5) signal peptidase complex subunit 3 from *D. reticulatus*. Similarly, through omics and in silico prediction algorithms, peptides from *O. moubata* midgut membrane antigens were selected for rabbit vaccination against *O. moubata* and *O. erraticus* [51]. Interestingly, the levels of protection were higher against *O. erraticus* than *O. moubata*. Other proteins in the Ixodidae and Argasidae families, such as *R. (B.) microplus* [46] and *O. erraticus* [47] aquaporins, *O. erraticus* ABC transporter, selenoproteins T, chitinase, secreted protein PK-4 [47,50], *Ixodes persulcatus* ferritin 2 [48] and *R. (B.) microplus* cystatins [49], showed limited cross-protection rates. Further studies are necessary to improve the protection levels achieved with these antigens.

## 4. Vaccines Blocking Pathogen Transmission

An alternative strategy to control tick-borne pathogens is through a vaccine that can block the transmission of these organisms to the host [81]. Some pathogens take advantage of tick saliva molecules for their transmission at an early stage of tick attachment, which increases their dissemination to the host [82]. If a vaccine is able to interfere in key tick physiology processes, pathogen transmission could be blocked. Examples of such processes and potential antigens include: (1) disruption of tick defense mechanisms targeting the microbiota biofilm modulator PIXR [83], (2) reduction in tick attachment time on the host by targeting the cement cone protein 64TRP [84,85], and (3) impairment of tick molecules that favor pathogen development and invasion into the tick, such as the antimicrobial protein IAFGP [86]. In this sense, a proposal for a universal anti-tick vaccine with a transmission-blocking potential is a promising tool to control ticks and pathogens, as suggested for Bm86 homologues to control *Borrelia burgdorferi* (Burgdorfer, 1982) host infection [87].

Another example of a candidate antigen to control pathogens includes a *R. (B.) microplus* subolesin fused with *Anaplasma marginale* (Theiler, 1910) Major Surface Protein 1a (SUB-MSP1a), which showed cross-protection potential against *R. (B.) annulatus* [88]. SUB-MSP1a was further used in bovine and sheep vaccination trials against *R. (B.) microplus* [89]. In this trial, the prevalence of *Babesia bigemina* (Wenyon, 1926) was reduced, showing the potential of SUB-MSP1a to control tick infestations and pathogen infection/transmission. Indeed, subolesin was also shown to be important for tick transmission of *Anaplasma phagocytophilum* (Foggie 1949), *A. marginale*, and *B. burgdorferi* [90]. Nevertheless, the multi-antigenic vaccine containing *R. (B.) microplus* subolesin, *R. appendiculatus* TRP64, and 3 histamine binding proteins, as well as *Theileria parva* (Theiler, 1904) sporozoite antigen p67C, did not affect *R. appendiculatus* survival and transmission of *T. parva* to cattle [91]. Using a different methodology, *I. ricinus* fed by artificial membranes with blood containing antibodies against an *I. scapularis* lipocalin or against an *I. scapularis* lectin, impaired tick engorgement, as well as *A. phagocytophilum* infection [92]. These two antigens identified using a vaccinomic approach are good examples of how this methodology can be useful for antigen discovery.

In addition to that, a live anti-tick vaccine was developed using attenuated *Babesia bovis* as a carrier to express tick proteins in calves [93]. The calves immunized with *B. bovis* carrying the *H. longicornis* GST gene generated anti-HlGST antibodies and developed mild babesiosis. *Rhipicephalus (B.) microplus* reared in these calves were smaller, and their fecundity was affected. This dual vaccine approach opens the perspective to develop universal vaccines not only against several tick species but also against tick-borne pathogens.

## 5. Advances and Remaining Challenges to Universal Tick Vaccines

This essay reviews studies showing tick antigens being evaluated for vaccine protection levels, including different stages of method development: from in silico or in vitro antigenicity, antibody cross-binding, and artificial feeding analysis, to vaccination trials using target hosts such as cattle, deer, and dogs. However, it is difficult at this point to compare them directly in order to find the best options to be included in a multi-antigenic vaccine. Even when limiting the comparison to antigens that were already analyzed in natural hosts, there are many biological variables to consider in differentiating vaccine efficacies, including host and tick diversity [39,94,95]. Moreover, although Canales’ method [33] for calculating the overall efficacy of the vaccine has become a standard and has been widely used in vaccination trials, alternative considerations have also been employed to evaluate levels of vaccine protection. These differences make it challenging to compare the level of protection across different studies [96,97]. The use of standard protocols would be helpful to allow the estimation of vaccine efficacy for ticks with similar biology and for similar experimental approaches.

Another important point in cross-vaccination studies is to verify if a given tick species is not actually two or more related but distinct species. Accurate species identification is fundamental to analyze homologous or heterologous vaccine efficacies [98]. Some examples of ticks that were the subject of vaccination trials and were under-speciated groups at the time include: (1) the formerly *Amblyomma cajenense* (Fabricius, 1787), now known as a complex of six valid species, including *A. sculptum* [99], (2) *R. sanguineus* s.l., which is contested to comprise multiple distinct species [100], and (3) *R. (B.) microplus*, which was also split into two species, giving rise to the new species *Rhipicephalus (Boophilus) australis* (Fuller, 1899) [101]. Thus, for example, vaccine antigen candidates such as protease inhibitors used against *A. sculptum* [102] must be considered as heterologous protection against other *A. cajennense* complex species. Importantly, the low protection rates obtained in some vaccination trials performed on *R. (B.) microplus* using TickGard and *R. australis* using Gavac could be the result of physiologic and/or Bm86 amino acid sequence variations between these two species [36,103,104]. Nevertheless, the selection of vaccine antigens from conserved proteins among tick species or isolates from different geographic areas will contribute to avoid vaccine inefficacy due to tick species misidentification [105,106]. In this context, the development of universal vaccines is highly desirable.

### 5.1. Rational Strategies for Improving Tick Vaccines

Different protocols for improving vaccines against multiple tick species were developed recently, thanks to advances in molecular biology, bioinformatics, and new material for vaccine components, such as adjuvants. Furthermore, the use of multiple antigens in a universal vaccine has been proposed, offering advantages such as protection against different tick species and different stages of the tick life cycle, in addition to the pathogens carried by them [107]. Multi-antigenic vaccines prove particularly advantageous for ticks with a single host, such as *R. (B.) microplus*, as they remain on the same host throughout their larval, nymphal, and adult stages. During this parasitic cycle, the tick produces distinct molecules to evade the host’s immune response [108,109,110,111]. Moreover, for multi-antigenic vaccines to be effective against ticks with three hosts, they must incorporate proteins specific to each life stage of the parasite that infests each host. Therefore, regardless of the number of hosts affected by tick infestations, a multi-antigenic vaccine should encompass protective antigens targeting various life stages of different tick species, as well as different secreted molecules by the same tick species. The use of multiple antigens in single vaccines or chimeric constructs has shown promising levels of protection in some cases, although some antigen combinations did not increase vaccine efficacy [107]. An interesting strategy proposed was the use of oral vaccines for the control of tick infestations, which, despite being incipient, affected tick survival and fertility, justifying further experiments [41,112].

In addition to in vivo trials, the immunogenicity of a number of tick molecules was analyzed in silico, revealing the potential of cross-protective tick antigens to be used for other tick species, such as aquaporin [113] and GSTs [114]. Similarly, potential proteins to be used for vaccines were discovered by reverse vaccinology analysis [97,115]. The use of systems biology is also a promising technique for the development of vaccines against ticks and their vectored pathogens [116]. Using this tool, it is possible to propose antigens for new generation vaccines that make use of transcriptomes, proteomes, and immunogenomics. Through “omics” analyses, it is possible to find not only antigens but also adjuvants for use in tick vaccines [107]. New generation adjuvants are under development to increase the immune response level, quality, and duration against infection and parasitic diseases [117,118]. For ticks, the performance of adjuvant formulations is currently being analyzed to improve the efficacy of known antigens such as Bm86, showing increased protections levels, although adverse reactions such as cutaneous side effects remain an issue [119]. A self-adjuvancing antigen carrier study indicated the immunogenicity, biodistribution, and safety of Bm86 carried by silica nanoparticles, demonstrating the potential of this delivery platform in anti-tick vaccine applications [120].

New vaccine delivery systems are being developed to protect against ticks and tick-borne diseases, including the use of lipid nanoparticle (LNP)-mRNA vaccines [121,122]. This technology offers an alternative to live attenuated or subunit-based vaccines [123]. An *Ixodes scapularis* salivary gland protein encoded by mRNA using this system decreased tick engorgement and *B. burgdorferi* transmission [122]. Interestingly, immunization with mRNA encapsulated into lipid nanoparticles induced not only a humoral immune response against Powassan virus, another pathogen transmitted by *I. scapularis*, but also against other tick-borne flaviviruses [121]. Other promising delivery systems include viral vectors [124] and live parasite vaccines [125].

### 5.2. Impact of Tick Expansion into New Territories for Vaccine Development

Tick geographic expansion is speeding up worldwide, including for species that affect human wealth or economic activities, such as *I. scapularis* [126], *H. longicornis* [127] and *R. (B.) microplus* [128,129]. The losses caused by tick parasitism worldwide are estimated to be around USD 20 to USD 30 billion annually [130], a figure that will likely increase as ticks move to new areas due to global climate change. The presence of compatible hosts, climate, physical, and biotic factors is essential for tick establishment into new territories [104,129,131,132], where tick-borne pathogens can eventually also be introduced. For long-distance dispersion, human action as well as mammal and bird migration play a main role in such expansions [133,134]. Otherwise, for shorter distances, factors such as proximity to a tick-parasitized area, forest cover, and rivers are predicted to be favorable for tick spread [126].

Appropriate climate is one major restriction for tick establishment into new areas [131]. Consequently, the ability to exploit new geographic areas relies on the tick ecological plasticity. Species Distribution Modeling (SDM) is an in silico methodology used to analyze climatic niche expansion of *I. ricinus*, the most important tick vector of pathogens in Europe [131]. This work predicted that *I. ricinus* is expanding to about two times the current area due to an increase in human demographics and CO_2_ production. Otherwise, changes in temperature, rainfall, and moisture can shrink or increase tick geographic range, as a model predicted for the eastern paralysis tick *Ixodes holocyclus* (Neumann, 1899) in Australia [132] or for the bont tick *Amblyomma hebraeum* (Koch, 1884) in Zimbabwe [135]. Similarly, *I. ricinus* expansion would be affected by the balance between rising temperatures that extends the questing season and favor survival, and droughts that promote tick mortality [136]. Model predictions were used to forecast suitable habitat for *R. (B.) microplus* distribution in West Africa [137] and South America [129], where temperature was a key determinant for potential tick invasion.

The presence of ticks in new geographic areas imposes challenges to control strategies since new hosts will be susceptible to ticks and tick-borne pathogens [138]. This is a concern, especially for highly generalist ticks such as *H. longicornis*, already spread in North America and with the potential for expansion to new areas in Russia, equatorial Papua New Guinea, and Pacific Islands [129]. Another example includes the wild host reservoir for pathogens, e.g., *B. burgdorferi* transmitted by *I. scapularis*, which could increase the difficulty to eliminate Lyme disease in North America [126]. In these cases, the success of vaccination programs could be hindered by the presence of wild animals living near humans and livestock. Additionally, the invasion of *R. (B.) microplus* into new areas can disturb not only hosts but also native tick species, as is the case of *R. (B.) decoloratus* in Angola [128], Benin [138], and South Africa [139]. The recent expansion and establishment of *R. (B.) microplus* in these countries has already resulted in the partial displacement of *R. (B.) decoloratus*, affecting the stability of native and non-native pathogens.

Monitoring new emergence, as well as predicting the invasion of ticks into specific areas, is essential to guide strategies to develop a vaccine protecting against several tick species. For instance, hosts such as bovines or white-tailed deer would be immunized, respectively, against *R. (B.) microplus* and *R. appendiculatus* in Africa [128,139], and *R. (B.) annulatus* and *R. (B.) microplus* in America [140]. Universal vaccines would be extremely valuable for regions where the economy is heavily dependent on livestock activity.

## 6. Future Directions

Although several antigens have shown promise to achieve anti-tick vaccines, there are many factors that present challenges in commercializing these vaccines [4]. Among these factors are the importance of the global market for acaricides and repellents, the safety and cost effectiveness of vaccines, as well as the need to protect against multiple ticks, pathogens, or other parasites in animals [4]. In this regard, cross-protection vaccines for specific ticks and tick-vector pathogens occupying the same geographic regions are highly dependent on antigen selection (Table 2). Moreover, most data obtained until now show enhancement of vaccine efficacy if the vaccine formulation includes proteins that are present in multiple structures or physiological processes of ticks [141]. The advances in vaccinology using genomic, transcriptomic, proteomic, RNA interference, phage display, DNA vaccine, epitope prediction, and other technologies have soared in recent years, and their contributions to finding new antigen candidates against multiple tick species is unquestionable [47,93,130,142,143,144,145]. Even for a well-known tick antigen such as Bm86, improved omics analyses and systems biology applications have been useful in expanding their use against different tick species [146]. For example, extensive functional polymorphism analysis of Bm86 from *R. (B.) microplus* endemic to different geographical regions indicates a cross-protection potential of this protein [147]. Effectively, methods to predict new B-cell epitopes using tick proteomes are under development [148], ranking antigenic peptides under different experimental conditions using structural information and sequence similarity. With this approach, fifteen complete sequences of *R. (B.) microplus* Bm86 were used to predict epitope immunogenic scores, and the most promising peptides can be tested in trials in order to select an antigen capable of protecting against multiple tick species [148].

The Bm86 antigen and their orthologs have been investigated for more than 30 years, and these studies support the idea of a universal vaccine, or at least a broad-spectrum vaccine, especially to protect against *R. (B.) annulatus* and *R. (B.) decoloratus* that coexist with *R. (B.) microplus* in America and Africa, respectively. However, despite developing cross-protection in most cases, vaccination with Bm86 and its orthologues does not always result in protection against different tick species [149]. Similarly, subolesin and its orthologs are emerging as promising candidates, since these proteins have been extensively tested against many tick species in laboratory and wild animals, including deer species endemic to America and Eurasia, which are involved in the maintenance of tick and tick-borne pathogen populations. A tick research consortium named CATVAC (Cattle Tick Vaccine Consortium) is working to develop a vaccine against *R. (B.) microplus* using some of the most promising antigens, including subolesin, to be commercialized in Africa [141]. Other tick research consortia for the development of anti-tick vaccines include ANTIDotE (Anti-tick Vaccines to Prevent Tick-borne Diseases in Europe), aiming to protect hosts against *I. ricinus* in Europe [150], and INCOGARR (Immunogens compatible with integrated management strategies in tick control), to protect ruminants against *R. (B.) microplus* using the P0 antigen [151]. The financing of such consortia combined with methodological advances, new vaccination protocols, and the use of multi-antigenic formulations, will bring anti-tick vaccines forward as desirable alternatives to control ticks and tick-borne pathogens in the near future.

## Figures and Tables

**Table 1 animals-13-02031-t001:** Vaccination trials showing cross-protective vaccine efficacies using tick antigen candidates.

Name	Target Tick Species	Cross-Protected Tick Species/Host	Vaccine Efficacy/Infestation Stage	References
Bm86 and orthologs	*R. (B.) microplus*	*Hyalomma anatolicum anatolicum* (Koch, 1844)/cattle	36.5% (L)	[36]
	*H. a. anatolicum*	*R. (B.) microplus*/cattle	26.8% (L); 25.1% (AF)	[36]
	*R. (B.) microplus*	*Hyalomma dromedarii* (Koch, 1844)/cattle	89% ^a^, 98% ^a^ (L); 29% ^a^, 36% ^a^ (AF)	[26]
	*R. (B.) microplus*	*H. dromedarii*/camel	26.7% ^a^, 31.3% ^a^ 38.6% ^a^ (AF)	[26]
Subolesin and orthologs	*Ornithodoros erraticus* (Lucas, 1849) ^d^	*Ornithodoros moubata* (Murray, 1877)/rabbit	~40.0% (L, N, AF)	[37]
	*O. moubata* ^d^	*O. erraticus*/rabbit	50.3% (L, N, AF)	[37]
	*H. a. anatolicum*	*R. (B.) microplus*/cattle	54% (L)	[38]
	*Rhipicephalus appendiculatus* (Neumann, 1901)	*Amblyomma variegatum* (Fabricius, 1794)/cattle ^b^	50% and 89% (L)	[39]
	*R. appendiculatus*	*Rhipicephalus (Boophilus) decoloratus* (Koch, 1844)/cattle ^c^	51% (L)	[39]
	*A*. *variegatum*	*R. appendiculatus*/cattle ^b^	86% and 83% (L)	[39]
	*A. variegatum*	*R. (B.) decoloratus*/cattle ^c^	72% (L)	[39]
	*R. (B.) decoloratus*	*R. appendiculatus*/cattle ^b^	66% and 89% (L)	[39]
	*R. (B.) decoloratus*	*A. variegatum*/cattle ^b^	58% and 94% (L)	[39]
	*Rhipicephalus haemaphysaloides* (Supino, 1897)	*Haemaphysalis longicornis* (Neumann, 1901)	79.3% and 86.6% (AF)	[40]
	*R. (B.) decoloratus*	*R. appendiculatus*/cattle	99% (L, N, AF)	[41]
Calreticulin	*H. a. anatolicum*	*R. (B.) microplus*/cattle	37.56% (L)	[38]
CathL	*H. a. anatolicum*	*R. (B.) microplus*/cattle	22.21% (L)	[38]
GST	*H. longicornis*	*R. appendiculatus*/rabbit	67% (AF)	[42]
	*R. (B.) decoloratus*/*A. variegatum* ^e^	*Rhipicephalus sanguineus* s.l. (Latreille, 1806)/rabbit	35% (AF)	[43]
P0	*R. sanguineus* s.l./*R. (B.) microplus* ^d^	*R. (B.) microplus*/cattle	96% (L)	[44]
	*R. sanguineus* s.l./*R. (B.) microplus* ^d^	*Amblyomma mixtum* (Koch, 1844)/rabbit	54% (L)	[45]
Aquaporin	*R. (B.) microplus*	*R. sanguineus* s.l./dog	7.2% (L); 4.5% (N)	[46]
	*O. erraticus*	*O. moubata*/rabbit	9.3% (AF)	[47]
ABC transporter	*O. erraticus*	*O. moubata*/rabbit	26.7% (AF); 15.4% (AM)	[47]
Selenoprotein T	*O. erraticus*	*O. moubata*/rabbit	18.6% (AM)	[47]
Ferritin	*Ixodes persulcatus* (Schulze, 1930)	*Ixodes ovatus* (Neumann, 1899)/guinea pig	~40% (AF)	[48]
Cystatin	*R. (B.) microplus*	*R. appendiculatus*/rabbit	11.5% (AF)	[49]
Chitinase	*O. erraticus*	*O. moubata*/rabbit	19.6% (AF)	[50]
Secreted protein PK-4	*O. erraticus*	*O. moubata*/rabbit	8.1% (N, AM)	[50]
OM03, OM85 and OM99 peptides	*O. moubata*	*O. erraticus*/rabbit	20.7% to 66.1% (N, AF, AM)	[51]

CathL, cathepsin L-like cysteine proteinase; GST, glutathione S-transferase; P0, 60S acidic ribosomal protein. L, larva; N, nymph; AF, adult female; AM, adult male. ^a^ protection for specific parameters. ^b^ For *Bos indicus* (Linnaeus, 1758) and *B. indicus* × *Bos taurus* (Linnaeus, 1758) breeds, respectively. ^c^
*B. indicus* × *B. taurus* breed. ^d^ peptide conjugates. ^e^ cocktail vaccine.

**Table 2 animals-13-02031-t002:** Cross-protection antigen candidates to control ticks and pathogens occupying the same geographic regions.

Geographic Region	Antigen	Target Tick Species	Target Tick Borne Disease	References
Africa and Asia	Bm86 and orthologs	*R. (B.) microplus*, *H*. *a. anatolicum*, *H. dromedarii*	*B. burgdorferi*	[26,36,53,87]
	Calreticulin and CathL	*R. (B.) microplus*, *H. a.anatolicum*		[38]
Africa	Subolesin and orthologs	*O. moubata*, *O. erraticus*, *R. (B.) microplus*, *H*. *a. anatolicum*, *A*. *variegatum*, *R. (B.) decoloratus*, *R. appendiculatus*		[37,39,41]
	ABC transporter	*O. moubata*, *O. erraticus*		[47]
	GST	*R. appendiculatus*, *R. (B.) decoloratus*, *A. variegatum*, *Rhipicephalus sanguineus* s.l.	*B. bovis*	[42,43,93]
	Aquaporin	*R. sanguineus* s.l., *O. moubata*		[46,47]
	Selenoprotein T, Chitinase, Secreted protein PK-4, OM03, OM85 and OM99 peptides	*O. moubata*, *O. erraticus*		[47,50,51]
Asia	Subolesin and orthologs	*R. (B.) microplus*, *H. anatolicum*, *H. longicornis*, *R. haemaphysaloides*	*B. burgdorferi*	[38,40,87]
	Ferritin	*I. ovatus*, *I. persulcatus*		[48]
Americas	Subolesin and orthologs	*R. (B.) microplus*, *R. (B.) annulatus*	*B. bigemina*	[89]
	P0	*R. (B.) microplus*, *A. mixtum*		[44,45]

CathL, cathepsin L-like cysteine proteinase; GST, glutathione S-transferase; P0, 60S acidic ribosomal protein.

## Data Availability

Not applicable.

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
