# Peer review of "Universal Tick Vaccines: Candidates and Remaining Challenges"

_animals, 2023, doi:10.3390/ani13122031_

Round 1

Reviewer 1 Report

This review is excellent. I just have some technical corrections regarding the citation of scientific names. I don't know if it's a journal rule, but scientific names should not be separated.

I recommend this review to be published.

Author Response

Reviewer: 1

Comments:

This review is excellent. I just have some technical corrections regarding the citation of scientific names. I don't know if it's a journal rule, but scientific names should not be separated.

Answer: Thank you for your suggestions. The manuscript was changed accordingly.

Reviewer 2 Report

Review of Universal tick vaccines.

The manuscript provides a useful essay on the concept of multi-target i.e. “universal” vaccines.

I found it to be interesting, informative, the product of expertise, and mostly thoughtful. I was not particularly convinced that this approach is the best, opposed to individually targeted vaccines. Nevertheless I recommend publication with some refinements for the consideration of the authors.

1. There is a statement at line 370 that there is evidence that multivalent vaccines are more efficacious (true) but that multi-targeted vaccines are more efficacious. I see no evidence for that and really that is counter-intuitive. It would certainly be desirable to have a vaccine effective against multiple species, but one is more likely to sacrifice some degree of efficacy. Please expand and explain.

2. There is missing discussion on an important aspect to this, and that is the difference between 3-host ticks and 1-host ticks in terms of escaping the host immune response. The 1-host tick strategy is to vary antigens that are exposed to the host during the 3 feeding stages, larva, nymph and adult, so that immune response against the larva, for example, is useless against the nymph etc.  Three host ticks don’t have to worry about that as much. To make a universal vaccine then, one must find antigens that are shared across species and life stages. Why no discussion of this important concept. Seems like an oversight.

3. There is a lot of discussion of Bm86 and its success without ever pointing out that it is a structural protein, unlike the other candidate antigens, and also a “hidden” antigen. Seems like an oversight that these points were not discussed.

4. The authors make a very important and valid point on the issue with target species identification (lines 270-285). In fact, it is interesting that Bm86 antigen was named for its discovery in the tick Boophilus microplus in the year 1986, which is now Boophilus australis, and perhaps explains why the Bm86 is not particularly effective against B. microplus. For that very same reason I would urge the authors to be more exact in their taxonomy. The author names should be appended to all scientific names at first mention in the manuscript.  Also the authors should be more consistent in distinguishing Rhipicephalus from Boophilus, whether they are considered to be full genera or subgenera, they are separate phylogenetic lineages. Thus, microplus, annulatus and decoloratus should be cited as Boophilus microplus, or if one prefers, as R. (B.) microplus.   Sanguineus can be cited as Rhipicephalus sanguineus or as R. (R.) sanguineus.  The distinction is all the more important because all Boophilus are one-host ticks with the immune escape clauses, where as none of the Rhipicephalus are one host ticks.

5. At line 119 the authors discuss a critical issue with variation in amino acid sequence being a source of differences in efficacy (protection) it should be noted that this variation within populations is the source for development of resistance to vaccines.

6. At line 351 there is some confounding of the alternative sylvatic host issues. They are very different in the two examples cited: Babesia in Boophilus and Borrellia in Ixodes. They should be discussed separately. In part this relates to the 3-host vs 1 host problem more than it does to sylvatic vs domestic host alternatives.

7. At line 309 the authors speak of “skin thickness” as an adverse reaction. Please clarify this remark. Are we talking hide damage or adverse injection site reaction (swelling)?

8. At lines 271 and 284 the term “speciation” is used where the authors must mean “species identification.”

9. At line 310 the authors use the word “adjuvanting” which is a common misspelling of “adjuvancing.”

10. At line 71, suggest omitting the word “high” and insert “a boost in”

11. At line 99, suggest changing word “induction” to “induces”

12. At line 126, Suggest after words “degradation machinery” add in parentheses “(proteases)”

13. For Table 1, column heading that says “Tick Species” change to “target species” or “target tick species.”

14. Summary, at line 22 there is awkward wording. Suggest omitting words “making being urgently” and inserting “creating urgency”

15. Line 258 has awkward wording. Perhaps it should say, “This essay reviews…”

16.  Line 265.  This sentence is so awkward I don’t even understand what the authors are trying to say. Please rewrite.

 Minor editing of English language required

Author Response

Reviewer: 2

Thank you very much for your comments. The manuscript was edited for proper English language and grammar.

Comments:

Nevertheless, I recommend publication with some refinements for the consideration of the authors.

Answer: Thank you very much for all your observations.

  1. There is a statement at line 370 that there is evidence that multivalent vaccines are more efficacious (true) but that multitargeted vaccines are more efficacious. I see no evidence for that and really that is counter-intuitive. It would certainly be desirable to have a vaccine effective against multiple species, but one is more likely to sacrifice some degree of efficacy. Please expand and explain.

Answer: We appreciate this reviewer’s suggestion. Please, note that in this sentence the “multi-targets” is related to various proteins in the same tick (the vaccine induces antibodies against different structures of the tick). To avoid any misunderstand (e.g. a vaccine against different tick species), we modified the text to clarify this question.

  1. There is missing discussion on an important aspect to this, and that is the difference between 3-host ticks and 1-host ticks in terms of escaping the host immune response. The 1-host tick strategy is to vary antigens that are exposed to the host during the 3 feeding stages, larva, nymph and adult, so that immune response against the larva, for example, is useless against the nymph etc. Three host ticks don’t have to worry about that as much. To make a universal vaccine then, one must find antigens that are shared across species and life stages. Why no discussion of this important concept. Seems like an oversight.

Answer: We appreciate this reviewer’s suggestion. Based on reviewer suggestion, we modified the manuscript to include this issue.

  1. There is a lot of discussion of Bm86 and its success without ever pointing out that it is a structural protein, unlike the other candidate antigens, and also a “hidden” antigen. Seems like an oversight that these points were not discussed.

Answer: Based on reviewer suggestion, we added more information about Bm86.

  1. The authors make a very important and valid point on the issue with target species identification (lines 270-285). In fact, it is interesting that Bm86 antigen was named for its discovery in the tick Boophilus microplus in the year 1986, which is now Boophilus australis, and perhaps explains why the Bm86 is not particularly effective against B. microplus. For that very same reason I would urge the authors to be more exact in their taxonomy. The author names should be appended to all scientific names at first mention in the manuscript. Also the authors should be more consistent in distinguishing Rhipicephalus from Boophilus, whether they are considered to be full genera or subgenera, they are separate phylogenetic lineages. Thus, microplus, annulatus and decoloratus should be cited as Boophilus microplus, or if one prefers, as R. (B.) microplus. Sanguineus can be cited as Rhipicephalus sanguineus or as R. (R.) sanguineus. The distinction is all the more important because all Boophilus are one-host ticks with the immune escape clauses, where as none of the Rhipicephalus are one host ticks.

Answer: We understand the reviewer's concern about species nomenclature. Based in reviewer’s comments, we have revised all scientific names.

  1. At line 119 the authors discuss a critical issue with variation in amino acid sequence being a source of differences in efficacy (protection) it should be noted that this variation within populations is the source for development of resistance to vaccines.

Answer: Based on reviewer suggestion, we added information about Bm86 variability in different tick populations.

  1. At line 351 there is some confounding of the alternative sylvatic host issues. They are very different in the two examples cited: Babesia in Boophilus and Borrellia in Ixodes. They should be discussed separately. In part this relates to the 3-host vs 1 host problem more than it does to sylvatic vs domestic host alternatives.

Answer: We agree with reviewer’s comment, and the information about Babesia/R. microplus was removed.

  1. At line 309 the authors speak of “skin thickness” as an adverse reaction. Please clarify this remark. Are we talking hide damage or adverse injection site reaction (swelling)?

Answer: We modified the text to clarify this question

  1. At lines 271 and 284 the term “speciation” is used where the authors must mean “species identification.”

Answer: It was corrected.

  1. At line 310 the authors use the word “adjuvanting” which is a common misspelling of “adjuvancing.”

Answer: It was corrected.

  1. At line 71, suggest omitting the word “high” and insert “a boostin”

Answer: It was corrected.

  1. At line 99, suggest changing word “induction” to “induces”

Answer: The sentence has been refined to ensure clarity and precision. (line 95)

  1. At line 126, Suggest after words “degradation machinery” add in parentheses “(proteases)”

Answer: The term "degradation machinery" was previously mentioned in the same paragraph, so, we include the word "proteases" near to the first citation.

  1. For Table 1, column heading that says “Tick Species” change to “target species” or “target tick species.”

Answer: It was corrected.

  1. Summary, at line 22 there is awkward wording. Suggest omitting words “making being urgently” and inserting “creating urgency”

Answer: It was corrected.

  1. Line 258 has awkward wording. Perhaps it should say, “This essay reviews…”

Answer: It was corrected.

  1. Line 265. This sentence is so awkward I don’t even understand what the authors are trying to say. Please rewrite.

Answer: The sentence has been refined to ensure clarity and precision.

Reviewer 3 Report

The manuscript titled: “Universal tick vaccines: candidates and remaining challenges” aims to summarize current knowledge regarding anti-tick vaccination, the challenges in its development and future directions towards successful implementation. The subject is interesting and fits the scope of the journal. However, at least two other comprehensive reviews of the same subject have been published in the last year, and this paper does not add a new angle to the subject. The writing of this manuscript is mostly fluent, but do not go in-depth in most subjects, and the organization of the manuscript is unclear. Moreover, the section regarding the epizootic spread of ticks and tick-borne diseases, although interesting and important, is not relevant to the subject of this review.

Main comments

-          More information should be given regarding the mechanism of action of each vaccine candidate.

-          It is sometimes unclear at each stage each vaccine is at. For example, the authors mention the commercial names for two Bm86, but do not state if these are commercially available and applied in the field (and what is the efficacy, and where in the world).

-          The authors mention the challenges in developing the vaccines, but after reporting initial success in the use of some candidates, no reason is given why these have not become commercially available.

-          Towards the end of the manuscript the authors provide a table with vaccine combinations suitable for various geographical areas. Are these suggestions complied by the authors? Or are these initiatives that are currently being developed? Please clarify.

Minor corrections

-          Line 22 – Please rephrase to: “making it urgent” or “emphasizing the urgency”.

-           Line 42 – I suggest “affect animal welfare” instead of “interfere with animal grazing”, to make it amore general statement, and not refer only to grazing animal species.

-          Lines 47-49 – unclear, please rephrase.

-          Bm86 – Please add a few sentences describing the protein and it function (similarly to the first few lines regarding Subolesin).

-          Lines 142-145 – Unclear. Please rephrase.

Author Response

Reviewer: 3

The manuscript titled: “Universal tick vaccines: candidates and remaining challenges” aims to summarize current knowledge regarding anti-tick vaccination, the challenges in its development and future directions towards successful implementation. The subject is interesting and fits the scope of the journal. However, at least two other comprehensive reviews of the same subject have been published in the last year, and this paper does not add a new angle to the subject. The writing of this manuscript is mostly fluent, but do not go in-depth in most subjects, and the organization of the manuscript is unclear.

Answer: We agree with the reviewer that the reviews have a partial overlap, however the reviews have different focuses. Our review has more detailed information and hypothesis about development and use a vaccine against various tick species. We think that the manuscripts are complementary and not redundant.

Moreover, the section regarding the epizootic spread of ticks and tick-borne diseases, although interesting and important, is not relevant to the subject of this review.

Answer: We understand the reviewer's concern about the relevance of the epizootic spread of ticks and tick-borne diseases. This topic was insert in the discussion because the challenge of tick control in new areas where there are new host and pathogens to be protected. In this sense, the development of a universal vaccine to include this new host and pathogens are high desirable. For this reason, we think this topic is very useful to compose a broad scenario where a universal vaccine against ticks is necessary.

More information should be given regarding the mechanism of action of each vaccine candidate.

Answer: We understand the reviewer's concern about mechanism of action of the vaccines. However, there is few information about the mechanism of action of tick vaccines. Basically, Bm86 vaccine is the unique vaccine that there is information about the biological mechanism, the antibodies against Bm86 induce lyses of gut cell. But even in this case, there is not a detail about this mechanism.

- It is sometimes unclear at each stage each vaccine is at. For example, the authors mention the commercial names for two Bm86, but do not state if these are commercially available and applied in the field (and what is the efficacy, and where in the world).

Answer: Based on reviewer suggestion, we added information about Bm86 vaccine details.  Commercial vaccines are not produced for other tick antigens.

- The authors mention the challenges in developing the vaccines, but after reporting initial success in the use of some candidates, no reason is given why these have not become commercially available.

Answer:  Thank you for your suggestions. This subject was added in the text accordingly.

- Towards the end of the manuscript the authors provide a table with vaccine combinations suitable for various geographical areas. Are these suggestions complied by the authors? Or are these initiatives that are currently being developed? Please clarify.

Answer: Thank you for your observation. The data showed in the table was compiled from published papers. The last column shows the references used to obtain the data.

Minor corrections

- Line 22 – Please rephrase to: “making it urgent” or “emphasizing the urgency”.

Answer: The sentence was also modified based on reviewer 2´s suggestions.

- Line 42 – I suggest “affect animal welfare” instead of “interfere with animal grazing”, to make it a more general statement, and not refer only to grazing animal species.

Answer: The sentence was changed according.

- Lines 47-49 – unclear, please rephrase.

Answer: The sentence was changed according.

- Bm86 – Please add a few sentences describing the protein and it function (similarly to the first few lines regarding Subolesin).

Answer: The sentence was also modified based on reviewer 2´s suggestions.

- Lines 142-145 – Unclear. Please rephrase.

Answer: The sentence was changed according